# Generation of stress fibers through myosin-driven reorganization of the actin cortex

Jaakko I Lehtimäki[1], Eeva Kaisa Rajakylä[2], Sari Tojkander[2], Pekka Lappalainen[1]*

[1]HiLIFE Institute of Biotechnology, University of Helsinki, Helsinki, Finland; [2]Section of Pathology, Department of Veterinary Biosciences, Faculty of Veterinary Medicine, University of Helsinki, Helsinki, Finland

**Abstract** Contractile actomyosin bundles, stress fibers, govern key cellular processes including migration, adhesion, and mechanosensing. Stress fibers are thus critical for developmental morphogenesis. The most prominent actomyosin bundles, ventral stress fibers, are generated through coalescence of pre-existing stress fiber precursors. However, whether stress fibers can assemble through other mechanisms has remained elusive. We report that stress fibers can also form without requirement of pre-existing actomyosin bundles. These structures, which we named cortical stress fibers, are embedded in the cell cortex and assemble preferentially underneath the nucleus. In this process, non-muscle myosin II pulses orchestrate the reorganization of cortical actin meshwork into regular bundles, which promote reinforcement of nascent focal adhesions, and subsequent stabilization of the cortical stress fibers. These results identify a new mechanism by which stress fibers can be generated *de novo* from the actin cortex and establish role for stochastic myosin pulses in the assembly of functional actomyosin bundles.

*For correspondence:
pekka.lappalainen@helsinki.fi

## Introduction

Cell migration, morphogenesis, and adhesion depend on contractile networks composed of actin and non-muscle myosin II (NMII) filaments. The forces in these structures are generated through sliding of bipolar NMII filaments along actin. Depending on the type of NMII isoform as well as on the organization and dynamics of actin filaments, different types of functional contractile actomyosin arrays can be generated in eukaryotic cells (*Lehtimäki et al., 2017a*). Assembly and contractility of these actomyosin arrays are controlled by upstream signals including Rho-family GTPases, kinase-phosphatase pathways, and $Ca^{2+}$ influxes (*Vicente-Manzanares et al., 2009*; *Prager-Khoutorsky et al., 2011*; *Tojkander et al., 2015*; *Tojkander et al., 2018*; *Burridge and Guilluy, 2016*).

Beneath the plasma membrane of animal cells lies a thin meshwork that is composed of actin filaments, NMII filaments, and associated proteins. This structure, the actin cortex, contributes to morphogenesis of interphase cells, and drives cell rounding during cytokinesis (*Bray and White, 1988*; *Chugh and Paluch, 2018*). In human cells, the assembly of actin cortex depends on both formins and the Arp2/3 complex (*Bovellan et al., 2014*; *Lu et al., 2017*; *Cao et al., 2020*). Ezrin, radixin, moesin (ERM)-family proteins link the cortical actin meshwork to the plasma membrane in interphase cells and during mitosis (*Bretscher et al., 2002*). Cortical actin meshwork is constantly under isotropic tension due to intracellular hydrostatic pressure and NMII-generated contractile forces to the actin cortex, and this can lead to formation of membrane blebs through local detachment of the actin cortex from the plasma membrane (*Charras and Paluch, 2008*). In a three-dimensional environment, some cell types can polarize and persistently migrate by stabilizing the bleb expansion

through rearward cortical flows and modulating the cortical tension (*Logue et al., 2015*; *Ruprecht et al., 2015*).

Whereas the actin cortex is composed of an irregular meshwork of actin filaments, animal cells also harbor highly ordered actomyosin structures. In many interphase cells, the most prominent actomyosin structures are called stress fibers. These thick, contractile actin bundles often connect to focal adhesions at their ends, and they are especially prominent in cells plated on stiff matrix or when external force is applied to the cells. Thus, stress fibers, together with focal adhesions, constitute a major mechanosensitive machinery in cells (*Tojkander et al., 2012*; *Livne and Geiger, 2016*). Apart from adhesion and mechanosensing, stress fibers contribute to cell morphogenesis and tail retraction during migration. Stress fibers also serve as precursors for sarcomeres during cardiomyocyte myofibrillogenesis (*Fenix et al., 2018*), and stress fiber-like actomyosin bundles contribute to interactions of epithelial cells with their neighbors and with basal lamina (*Yamada and Nelson, 2007*; *Munjal et al., 2015*; *López-Gay et al., 2020*; *Rajakylä et al., 2020*).

Stress fibers can be classified into different sub-types based on their protein compositions and association with focal adhesions. *Ventral stress fibers* are thick actomyosin bundles that are connected from their both ends to focal adhesions at the bottom of the cell. Despite their name, the central regions of ventral stress fibers often rise toward the dorsal surface of cell (*Naumanen et al., 2008*; *Burnette et al., 2014*). In many cell types, ventral stress fibers associate with each other to form a complex, mechanically interconnected network (*Xu et al., 2012*; *Kassianidou et al., 2017*). *Dorsal stress fibers* (also known as radial fibers) are non-contractile actin filament bundles that are generated at the cell front through formin- and VASP-mediated actin filament assembly at focal adhesions (*Hotulainen and Lappalainen, 2006*; *Tee et al., 2015*; *Tojkander et al., 2015*). *Transverse arcs*, on the other hand, are thin, contractile actomyosin bundles, arising through NMII-promoted condensing of the lamellipodial actin network at the cell edge (*Hotulainen and Lappalainen, 2006*; *Burnette et al., 2011*; *Tojkander et al., 2011*). Following their appearance, transverse arcs undergo retrograde flow toward the cell center, fuse with each other into thicker bundles, and become more contractile due to concatenation and persistent expansion of NMII filaments into stacks (*Tojkander et al., 2015*; *Fenix et al., 2016*; *Beach et al., 2017*; *Hu et al., 2017*; *Jiu et al., 2019*). Although transverse arcs are not directly linked to focal adhesions, they associate with focal adhesion-connected dorsal stress fibers (*Burnette et al., 2014*; *Senger et al., 2019*). Ventral stress fibers can be generated from the transverse arc and dorsal stress fiber network through a complex process that involves both mechanosensitive regulation of actin filament assembly at focal adhesions, as well as inhibition of actin disassembly within the stress fiber network (*Hayakawa et al., 2011*; *Tojkander et al., 2015*; *Tojkander et al., 2018*; *Lee and Kumar, 2020*). Moreover, pre-existing ventral stress fibers can undergo 'splitting' to generate new, adjacent ventral stress fibers (*Young and Higgs, 2018*). However, whether stress fibers can also be generated through other mechanisms has remained elusive.

Here we show that different cell types also exhibit, at their ventral actin cortex, thin stress fibers that are connected to focal adhesions at both ends. These actomyosin bundles form predominantly underneath the nucleus, and are less contractile and more dynamic compared to the ventral stress fibers, which are derived through fusion of transverse arcs at the lamella of migrating cells. Importantly, we demonstrate that assembly of these thin actomyosin bundles does not involve transverse arcs or any other stress fiber precursor. Instead, they are generated *de novo* from the actin cortex through NMIIA-driven reorganization of the actin filament meshwork, and hence we call them as cortical stress fibers.

## Results

### The actin cortex harbors cortical stress fibers of various size and orientation

Ventral stress fibers were originally defined as contractile actomyosin bundles, which attach to focal adhesions at their both ends (*Small et al., 1998*; *Hotulainen and Lappalainen, 2006*). However, migrating mesenchymal cells harbor ventral stress fibers of various length, orientation, and thickness, and these can either locate entirely at the ventral surface of cells or rise toward the dorsal surface from their central regions (*Prager-Khoutorsky et al., 2011*; *Burnette et al., 2014*; *Elkhatib et al.,*

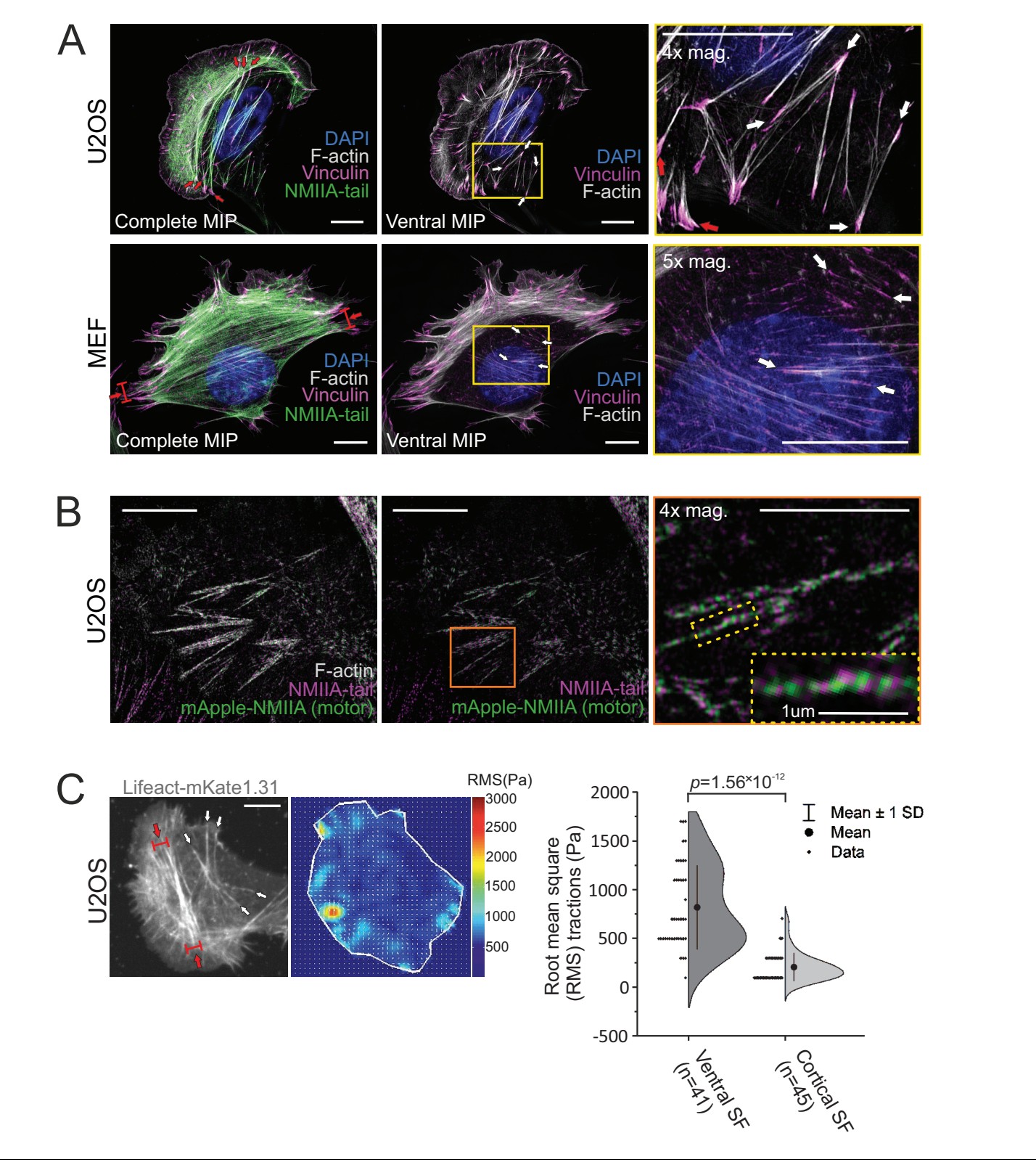

**Figure 1.** Stress fiber architecture of migrating cells. 3D-structured illumination microscopy (SIM) maximum intensity projections (MIPs) of the actomyosin networks in cells migrating on fibronectin. (**A**) Human osteosarcoma (U2OS) and mouse embryonic fibroblast (MEF) cells, where the panels on the left display complete MIPs. The panels in the middle show only the filament structures close to the ventral plane ('ventral MIP'), and the panels on the right are magnifications of the boxed regions from the middle panels. Red arrows highlight ventral stress fibers, and white arrows indicate

*Figure 1 continued on next page*

*Figure 1 continued*

examples of thin cortical stress fibers that are embedded at the cell cortex. DAPI (blue) and phalloidin (gray) were applied to mark the F-actin and nucleus, respectively. Vinculin (magenta) and non-muscle myosin II (NMII)A tails (green) were detected with respective specific antibodies. (B) 3D-SIM MIP projection from the ventral plane of a U2OS cell transfected with mApple-NMIIA construct (motor, green) and stained with NMIIA-tail specific antibody (magenta, tail) and phalloidin to visualize F-actin. 4× and 10× magnifications (orange box and yellow dotted box, respectively) show the bipolar NMIIA filaments in cortical stress fibers. (C) Traction force microscopy (TFM) analysis of the forces exerted by ventral stress fibers (red arrows) and cortical stress fibers (white arrows) to the underlying substrate. On the left, exemplary image of a LifeAct-mKate1.31-expressing U2OS cell and the obtained force map with root mean square tractions (RMS). Quantification of the RMS forces between the two stress fiber subtypes (n = 41 for ventral stress fibers [SF] and 45 for cortical stress fibers) is shown on right as half-violin plot including binned individual data points and mean with ±1 standard deviation (SD). p = 1.56 × 10$^{-12}$ (Mann–Whitney *U*-test, including outliers). Scale bars 10 μm and 5 μm for whole cell images and magnifications, respectively.

The online version of this article includes the following source data and figure supplement(s) for figure 1:

**Source data 1.** Combined data of RMS traction forces.
**Figure supplement 1.** Cortical stress fibers display periodic non-muscle myosin II (NMII) — α-actinin pattern and assemble on different ECMs.
**Figure supplement 2.** Cortical stress fibers are not linked to fibrillar adhesions.

*2014*; *Schulze et al., 2014*; *Baird et al., 2017*; *Lehtimäki et al., 2017b*; *Kumari et al., 2020*). To uncover the possible molecular differences between these diverse stress fibers, we utilized the 3D-structured illumination microscopy (SIM) on human osteosarcoma (U2OS) and mouse embryonic fibroblast (MEF) cells migrating on fibronectin. Consistent with previous literature, NMIIA containing, focal adhesion-attached stress fibers of varying thickness and length were visible in both cell lines (*Figure 1A*, white and red arrows). In addition to thick ventral stress fibers that connect focal adhesions located at the opposite sides of the cell, both cell types exhibited thin and relatively short actomyosin bundles that were connected to small focal adhesions at their both ends. As illustrated by the temporal-color coded 3D-SIM projections of F-actin, these thin actomyosin bundles reside at the immediate vicinity of the ventral cortex of the cell (*Figure 1—figure supplement 1A*, white arrows), whereas typical ventral stress fibers (*Burnette et al., 2014*; *Tojkander et al., 2015*) rise toward the dorsal surface from the middle of the bundle (*Figure 1A* and *Figure 1—figure supplement 1A*; red arrows). We named these thin, actin cortex-associated actomyosin bundles as *cortical stress fibers*.

Similar to the ventral stress fibers, NMII filaments displayed a bipolar arrangement in the small, basally located cortical stress fibers (*Figure 1B*). However, NMII did not appear to assemble into stacks, and the periodic pattern of actin filament cross-linking protein α-actinin was less regular in cortical stress fibers compared to transverse arcs and ventral stress fibers (*Figure 1—figure supplement 1B*). Moreover, the forces exerted by cortical stress fibers to the ECM were much smaller compared to typical ventral stress fibers (*Figure 1C*). Although cortical stress fibers were able to form also on laminin and collagen, they were most prevalently observed in cells migrating on fibronectin (*Figure 1* and *Figure 1—figure supplement 1C*). Due to their apparent fibronectin preference, we investigated the cortical stress fibers for fibrillar fibronectin deposits and Y118-phospho(p)-paxillin, two markers that are typical to fibrillar and focal adhesions, respectively (*Zaidel-Bar et al., 2007*; *Geiger and Yamada, 2011*). Apart from few adhesions in MEFs, cortical stress fibers did not associate with fibronectin deposits (*Figure 1—figure supplement 2A*). Moreover, the adhesions at the ends of cortical stress fibers contained Y118-phospho(p)-paxillin, indicating that they were not associated with fibrillar adhesions (*Figure 1—figure supplement 2B*). Taken together, the molecular composition of cortical stress fibers closely resembles the one of ventral stress fibers, but they are typically much smaller, localize consistently at the ventral actin cortex, and exert only weak traction forces.

## Cortical stress fibers are generated *de novo* from the ventral actin cortex

Ventral stress fibers are generated from a network of pre-existing transverse arcs and focal adhesion-attached dorsal stress fibers (*Tojkander et al., 2015*; *Tojkander et al., 2018*). Thus, we examined if the thin cortical stress fibers are generated by the same or a different mechanism. To this end, we imaged the ventral cortex of migrating U2OS cells and MEFs expressing LifeAct-TagGFP2 (to detect F-actin) and vinculin-mApple (to visualize focal adhesions) by time-lapse total internal reflection microscopy (TIRFM). Surprisingly, these experiments revealed that cortical stress fibers

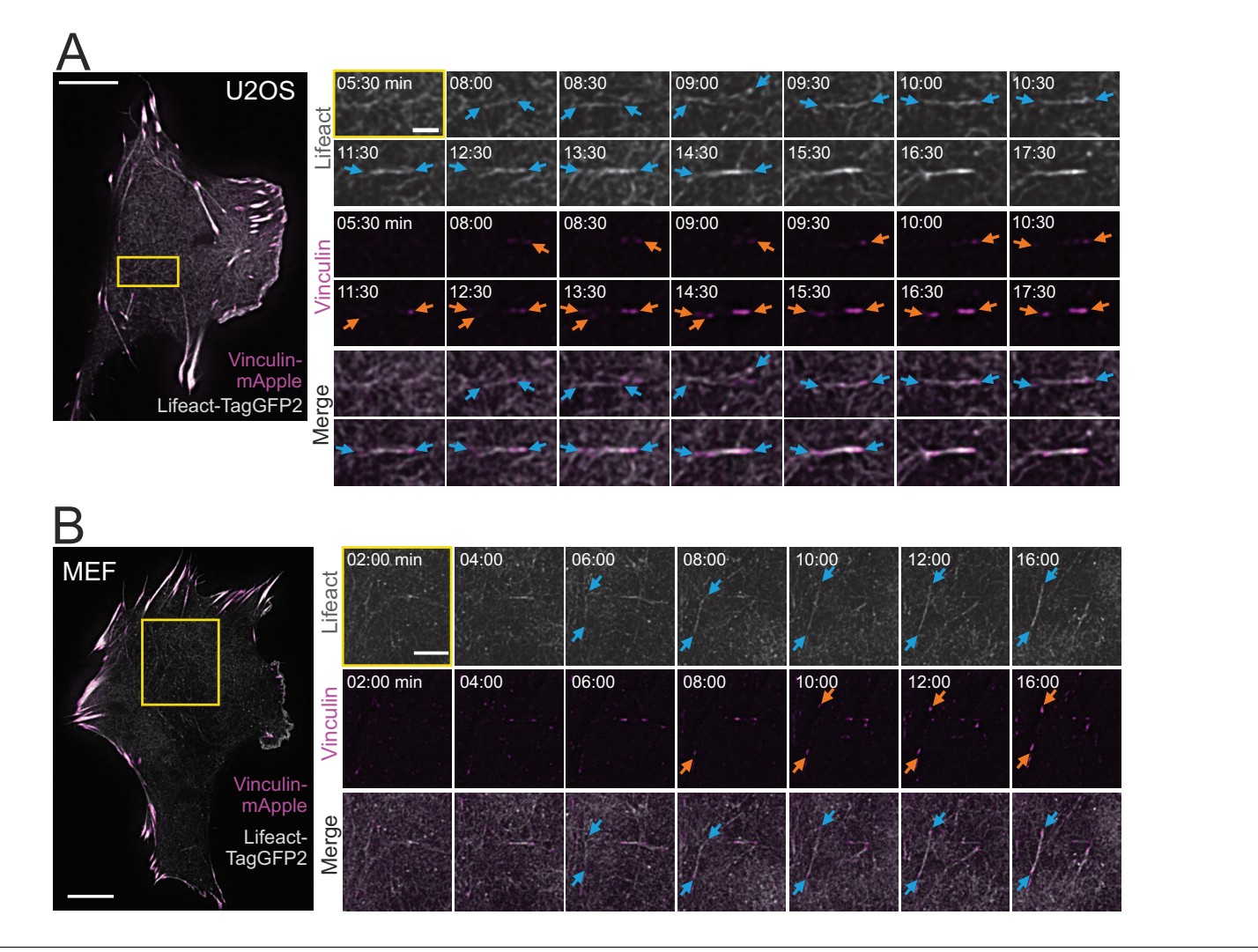

**Figure 2.** Cortical stress fibers assemble *de novo* from the actin cortex. TIRF time-lapse imaging of a migrating human osteosarcoma (U2OS) (**A**) and mouse embryonic fibroblast (MEF) cell (**B**) expressing LifeAct-TagGFP2 (gray) and vinculin-mApple (magenta). Selected time-lapse frames from the magnified areas (yellow box) are shown on the right as separate channels and merged frames. These demonstrate the *de novo* emergence of a cortical stress fiber from the ventral actin cortex. Blue arrows illustrate F-actin bundling and orange arrows point the maturation of the vinculin-positive focal adhesions. See also *Videos 1* and *2*. Scale bars 10 µm and 2 µm for whole cell images and time-lapse zoom-ins, respectively. Imaging interval 30 s. The online version of this article includes the following source data and figure supplement(s) for figure 2:

**Figure supplement 1.** Cortical stress fibers emerge from the ventral actin cortex through different intermediate assembly states.

**Figure supplement 1—source data 1.** PA-actin intensity profiles from photoactivated foci.

emerged *de novo* from the ventral actin cortex, without involvement of any pre-existing stress fiber precursors (*Figure 2A and B* and *Videos 1* and *2*). In this process, actin filaments of the cell cortex reorganized into thicker bundles (*Figure 2A and B*, blue arrows) and this was followed by growth of nascent, initially barely visible, vinculin-positive adhesions at the both ends of the bundle (*Figure 2A and B*, orange arrows). This eventually led to the formation of an actin filament bundle that was connected to focal adhesions at its both ends.

Because dorsal stress fibers elongate through actin polymerization at focal adhesions (*Hotulainen and Lappalainen, 2006*; *Tee et al., 2015*; *Tojkander et al., 2015*), we also examined the possible role of actin filament assembly at adhesions in the formation of cortical stress fibers. For this purpose, we photoactivated small regions of stress fibers adjacent to adhesions in U2OS cells expressing photoactivatable (PA) actin, and subsequently followed the movement of the

photoactivated actin foci. Whereas the photoactivated regions in dorsal stress fibers displayed a centripetal movement with a typical rate of ~0.3 µm/min, the photoactivated regions in cortical stress fibers were nearly immobile. This provides evidence that, unlike dorsal stress fibers, cortical stress fibers do not elongate from adhesions at their ends (*Figure 2—figure supplement 1A and B*).

Live-cell imaging experiments also demonstrated that cortical stress fibers are typically very dynamic with relatively short half-life (~25 min). However, it is important to note that the half-life of cortical stress fibers was highly variable from few minutes to more than 1.5 hr. Moreover, cortical stress fibers often associated dynamically with each other, and they could also arise via various intermediate assembly states (*Figure 2—figure supplement 1C* and *Video 3*). Collectively, these data reveal that cortical stress fibers are relatively dynamic actomyosin bundles, which assemble through a novel mechanism from the actin cortex, without involvement of stress fiber precursors or actin polymerization at focal adhesions.

## Cortical stress fibers preferably emerge underneath the nucleus in migrating cells

Cortical stress fibers were enriched at the rear of migrating cells and were typically located either underneath or close to the nucleus (*Figure 1*). Nucleus is a relatively bulky organelle, localized behind the lamella at the cell rear in polarized mesenchymal cells, and it undergoes NMII-dependent translocation as the cell moves forward (*Wu et al., 2014*; *Thomas et al., 2015*). Thus, we examined possible connection between the nucleus and generation of cortical stress fibers. Live-imaging of U2OS cells expressing Histone-H2B-mCherry to mark the nucleus, as well as LifeAct-TagGFP2 and focal adhesion marker miRFP670-paxillin, revealed that the assembly of cortical stress fibers occurred typically under the nucleus during cell locomotion (*Figure 3A* and *Video 4*). Similar results were obtained when nuclei were visualized together with miRFP670-paxillin and mApple-NMIIA (*Figure 3—figure supplement 1A* and *Video 5*). Quantification of the emergence of cortical stress

fibers from 19 time-lapse videos revealed that ~80% of the cortical stress fibers assembled underneath the nucleus (*Figure 3B*). By considering that the nucleus typically consists of <30% of the cell area, the assembly of cortical stress fibers occurs predominantly underneath the nucleus.

To elucidate if actin filament bundles resembling cortical stress fibers also exist in cells migrating in three-dimensional (3D) environment, we examined the distributions of adhesions, actin filaments, and nuclei in U2OS cells migrating in 3D fibrin network. In addition to thick, long actin filament bundles, which resemble 'ventral stress

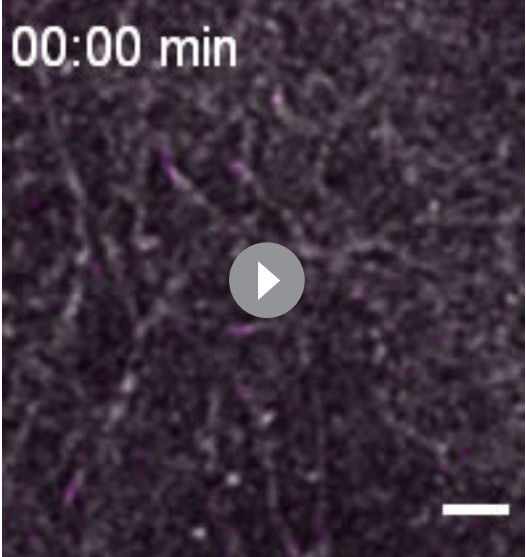

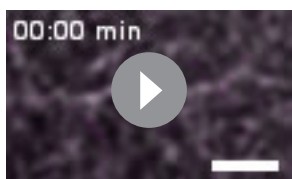

**Video 1.** Cortical stress fiber formation at the ventral actin cortex of a migrating human osteosarcoma (U2OS) cell. Zoom in of a U2OS cell expressing LifeAct-TagGFP2 (gray) and mApple-vinculin (magenta), migrating on a fibronectin (10 µg/ml)-coated high-precision imaging dish. Related to *Figure 2A*. Time-lapse ring-TIRFM video was recorded with Deltavision OMX SR with 30 s imaging interval. Playback rate 10 frames/s. Scale bar 2 µm.
https://elifesciences.org/articles/60710#video1

**Video 2.** Cortical stress fiber formation at the ventral actin cortex of a migrating mouse embryonic fibroblast (MEF) cell. Zoom in of a MEF cell expressing LifeAct-TagGFP2 (gray) and mApple-vinculin (magenta), migrating on a fibronectin (10 µg/ml)-coated high-precision imaging dish. Related to *Figure 2B*. Time-lapse ring-TIRFM video was recorded with Deltavision OMX SR with 30 s imaging interval. Playback rate 10 frames/s. Scale bar 2 µm.
https://elifesciences.org/articles/60710#video2

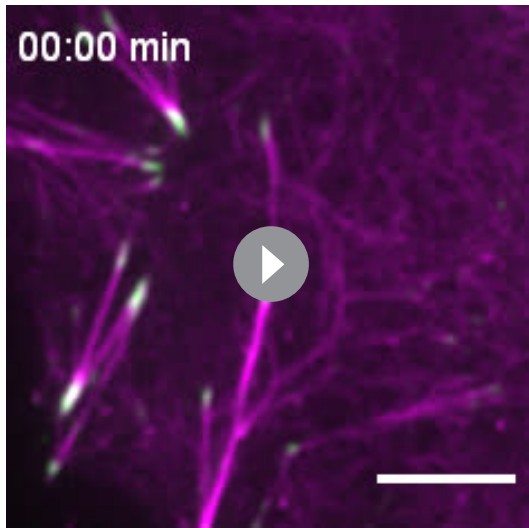

**Video 3.** Cortical stress fiber assembly at the trailing edge of a migrating human osteosarcoma (U2OS) cell. Zoom in of a U2OS cell expressing LifeAct-TagGFP2 (magenta) and mApple-vinculin (green), migrating on a fibronectin (10 µg/ml)-coated high-precision imaging dish. Related to *Figure 2—figure supplement 1C*. Time-lapse ring-TIRFM video was recorded with

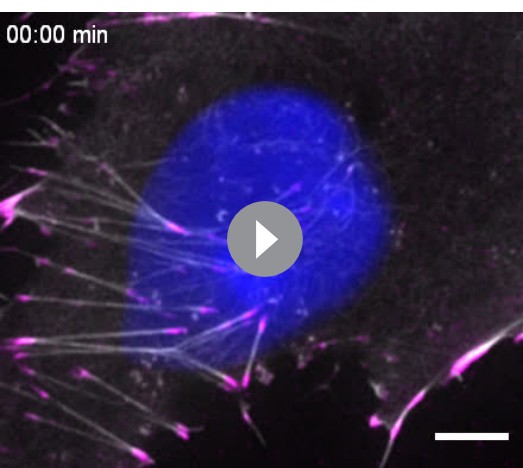

**Video 4.** Cortical stress fiber assembly under the nucleus of migrating human osteosarcoma (U2OS) cell. Zoom in of a U2OS cell expressing LifeAct-TagGFP2 (gray), histone-H2B-mCherry (blue), and miRFP670-paxillin (magenta), migrating on a fibronectin (10 µg/ml)-coated high-precision imaging dish. Related to *Figure 3A*. Time-lapse ring-TIRFM video was recorded with Deltavision OMX SR with 30 s imaging interval. Playback rate 20 frames/s. Scale bar 5 µm.
https://elifesciences.org/articles/60710#video4

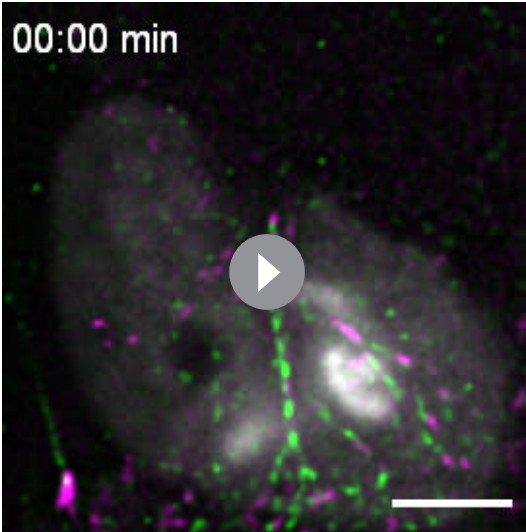

**Video 5.** Non-muscle myosin II (NMII)A pulses assemble into cortical stress fibers under the nucleus of migrating human osteosarcoma (U2OS) cell. Zoom in of a U2OS cell expressing histone H2B-eGFP (gray), mApple-NMIIA (green), and miRFP670-paxillin (magenta), migrating on a fibronectin (10 µg/ml)-coated high-precision imaging dish. Related to *Figure 3—figure supplement 1A*. Time-lapse ring-TIRFM video was recorded with Deltavision OMX SR with 20 s imaging interval. Playback rate 20 frames/s. Scale bar 5 µm.
https://elifesciences.org/articles/60710#video5

fibers' observed in cells migrating on 2D matrix, U2OS cells in 3D fibrin matrix also exhibited relatively short and thinner actin filament bundles, which terminated to paxillin-rich adhesions at their both ends (*Figure 3—figure supplement 1B* and *Video 6*). These filament bundles were prominent in the vicinity of the nuclei, suggesting that they may be related to cortical stress fibers.

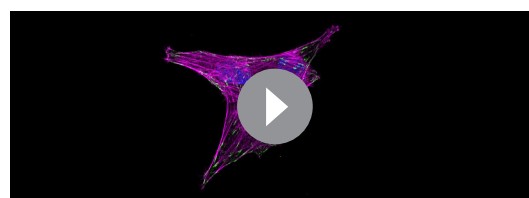

**Video 6.** 3D-reconstruction of human osteosarcoma (U2OS) cells migrating in a 3D fibrin matrix. U2OS cells expressing eGFP-paxillin (green), stained further with phalloidin for F-actin (magenta) and DAPI (blue) for visualization of the nuclei. Related to *Figure 3—figure supplement 1B*. Confocal stacks were acquired and deconvolved with Andor Dragonfly 505; 3D-reconstruction performed with Imaris 9.2.1. Scale bar size (µm) is proportional to the degree of zoom in the video and is indicated at the lower left corner.
https://elifesciences.org/articles/60710#video6

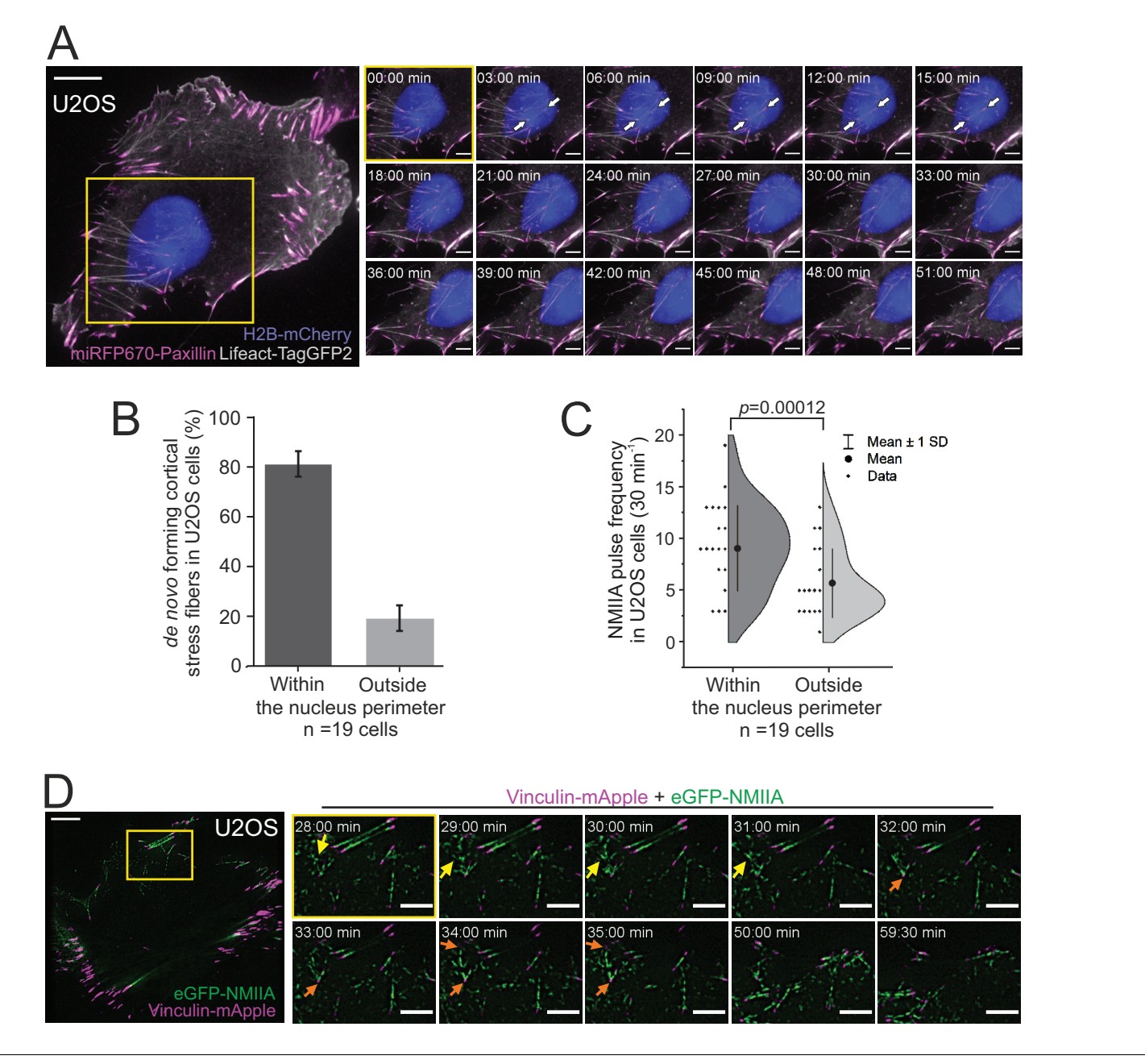

**Figure 3.** Non-muscle myosin II (NMII)A pulse-mediated cortical stress fibers assembly occurs predominantly underneath the nucleus in migrating cells. (A) TIRF time-lapse imaging of a migrating human osteosarcoma (U2OS) cell. The selected time-lapse frames (from the yellow boxed area) illustrate the assembly of a cortical stress fiber (exemplified by white arrows) below the moving nucleus. Histone H2B-mCherry was applied to detect the nucleus (blue, EPI-TIRF) and miRFP670-paxillin (magenta) and LifeAct-TagGFP2 (gray) to visualize focal adhesions and F-actin, respectively. See also *Video 4*. (B) Quantification of the position of *de novo* formation of cortical stress fibers from TIRF-time-lapse videos of U2OS cells expressing LifeAct-TagGFP2, Histone-H2B-mCherry, and miRFP670-paxillin. Data are presented as %-distribution of cortical stress fibers assembled under the nucleus versus outside the nucleus perimeter ± SEM. n = 101 cortical stress fibers analyzed from 19 cells. (C) Quantification of NMIIA pulse location at the ventral cortex of U2OS cells expressing Histone H2B-eGFP and mApple-NMIIA. Data are presented as pulse frequency (number of individual pulses) under the nucleus vs. outside the nucleus perimeter (rest of the ventral cortex). Each data point is normalized to the area of nucleus. Half-violin plot displays binned individual data points and mean with ±1 standard deviation (SD), n = 19 cells. Significance (p = 0.00012) tested with a paired *t*-test. (D) TIRF time-lapse imaging of a migrating U2OS cell expressing eGFP-NMIIA (green) and vinculin-mApple (magenta). Selected time-lapse frames from the magnified area (yellow box) demonstrate that NMIIA pulses (yellow arrows) are associated with the assembly of a cortical stress fiber and enforcement of vinculin positive focal adhesions (orange arrows). See also *Figure 4C* and *Video 7*. For (A) and (D), scale bars 10 µm and 5 µm for whole cell images and time-lapse zoom-ins, respectively. Imaging interval 30 s.

*Figure 3 continued on next page*

*Figure 3 continued*

The online version of this article includes the following source data and figure supplement(s) for figure 3:

**Source data 1.** Analysis of *de novo* forming cortical SFs.
**Source data 2.** Analysis of NMIIA pulse dynamics.
**Figure supplement 1.** Myosin pulses are associated with cortical stress fibers assembly underneath the nucleus, and morphologically similar adhesion-attached actin filament bundles also exist in cells migrating in 3D matrix.

## Myosin pulses direct cortical stress fiber assembly and maturation

Pulsatile behavior of NMII at the actin cortex has been documented in epithelial and mesenchymal cells, and reported to induce transient accumulation of cortical actin (*Kim and Davidson, 2011*; *Munjal et al., 2015*; *Baird et al., 2017*). Our analysis indicated that NMIIA pulses at the ventral cortex of U2OS cells are slightly more prevalent underneath the nucleus, where cortical stress fibers also principally assembled, compared to the region outside the nucleus perimeter (*Figure 3C*). To reveal the possible role of NMII pulses in generation of the cortical stress fibers, we performed TIRFM imaging of U2OS cells expressing eGFP-NMIIA and vinculin-mApple. Interestingly, assembly of cortical stress fibers and enforcement of focal adhesions at their ends were associated with NMIIA pulses (*Figure 3D* and *Video 7*). To examine this in more detail, we performed three-color TIRFM on U2OS cells expressing mApple-NMIIA, LifeAct-TagGFP2, and miRFP670-paxillin. These experiments revealed that NMIIA pulses frequently coincided with F-actin bundling (*Figure 4A* and *Video 8*). In majority of cases the NMIIA pulses were transient and 'non-productive', and the actin filament bundles collapsed back to the ventral cortex along with the cessation of the pulse (*Figure 4B*). However, in some cases the NMIIA pulses were more persistent and 'productive', and led to the formation of more stable actomyosin bundles. These actomyosin bundles engaged paxillin positive focal adhesions, causing them to align to the direction of the pull and to increase in size, thus leading to stabilization and maturation of the cortical stress fiber (*Figure 4A and C*). Importantly, focal adhesions associating with the new cortical stress fiber could either emerge *de novo* (*Figure 4—figure supplement 1A*, orange arrow, *Video 9*) or be pre-existing adhesions that were connected to other, discrete contractile bundles (*Figure 4—figure supplement 1B*, orange arrow and *Videos 8* and *10*). Moreover, we occasionally observed assembly of cortical stress fibers, where one end of the contractile bundle was connected to the mesh of pre-existing actomyosin bundles

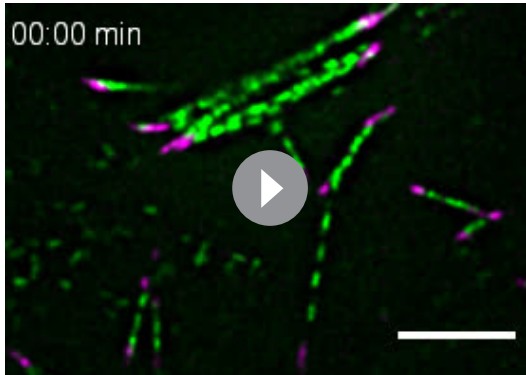

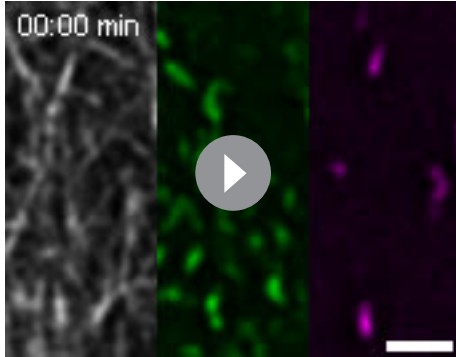

**Video 7.** Non-muscle myosin II (NMII)A pulses at the trailing edge of a migrating human osteosarcoma (U2OS) cells. Zoom in of a U2OS cell expressing eGFP-NMIIA (green) and mApple-vinculin (magenta) migrating on a fibronectin (10 µg/ml)-coated high-precision imaging dish. Related to *Figure 3D* and *Figure 4C*. Time-lapse ring-TIRFM video was recorded with Deltavision OMX SR with 30 s imaging interval. Playback rate 20 frames/s. Scale bar 5 µm.
https://elifesciences.org/articles/60710#video7

**Video 8.** Non-muscle myosin II (NMII)A pulses coordinate the cortical stress fiber assembly in a migrating human osteosarcoma (U2OS) cell. Zoom in of a U2OS cell expressing LifeAct-TagGFP2 (gray), mApple-NMIIA (green), and miRFP670-paxillin (magenta), migrating on a fibronectin (10 µg/ml)-coated high-precision imaging dish. Related to *Figure 4A*. Time-lapse ring-TIRFM video was recorded with Deltavision OMX SR with 30 s imaging interval. Playback rate 10 frames/s. Scale bar 2 µm.
https://elifesciences.org/articles/60710#video8

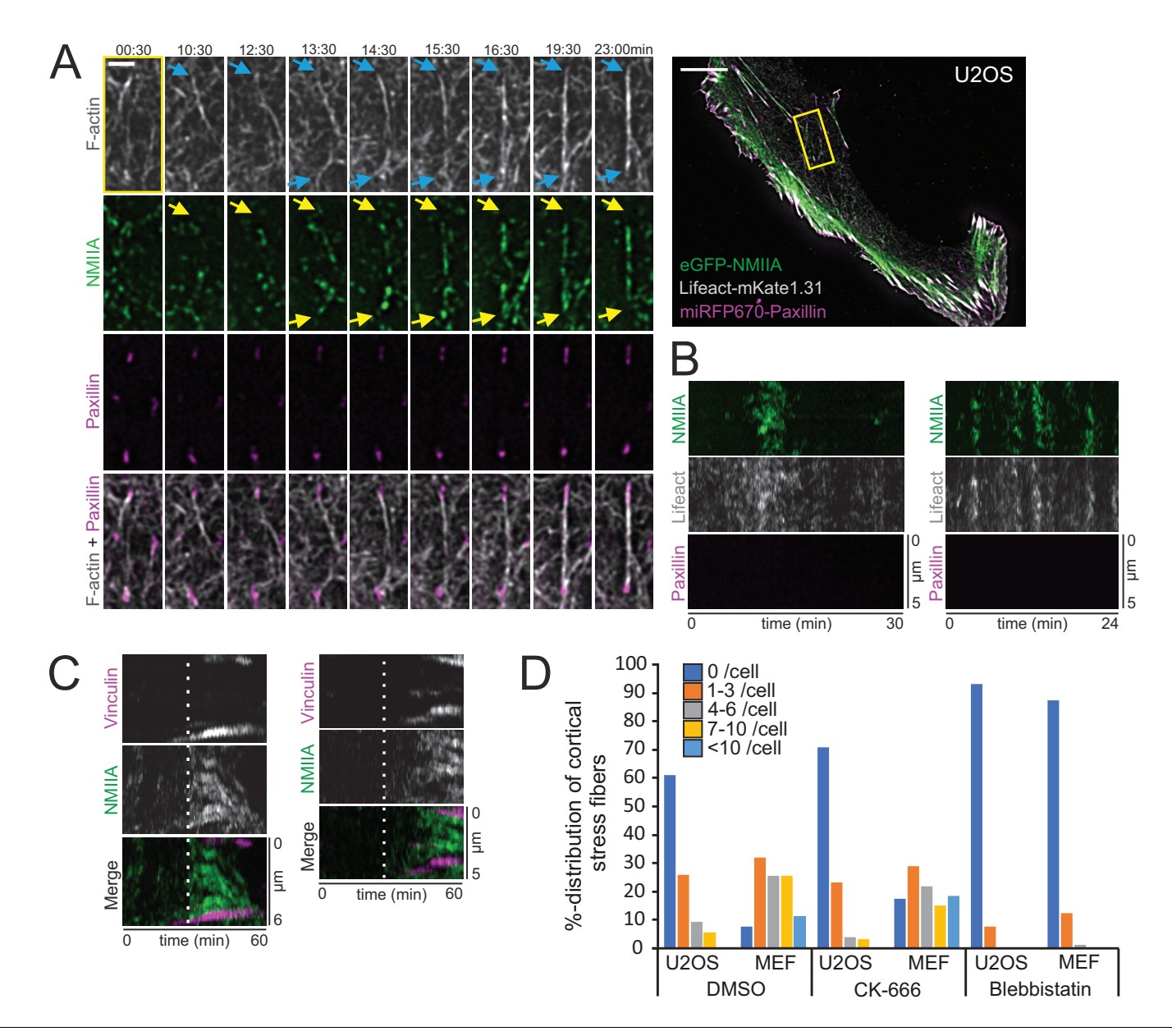

**Figure 4.** Cortical non-muscle myosin II (NMII) pulses orchestrate F-actin bundling for the assembly of cortical stress fibers. (**A**) Formation of a cortical stress fiber from the ventral cortex of a migrating human osteosarcoma (U2OS) cell as studied by time-lapse TIRF microscopy. The entire cell is shown on the right, and the selected time-lapse frames (from the region indicated by yellow box) display how F-actin (gray, LifeAct-mKate1.31; blue arrows) and NMIIA (green, eGFP-NMIIA; yellow arrows) organize into an actomyosin bundle, which promotes the enlargement of pre-existing paxillin-positive focal adhesions (FA) (magenta, miRFP670-paxillin). Note that prior to bundle assembly, the pre-existing focal adhesions were connected to different actomyosin bundles. Original imaging interval 30 s. See also *Video 8*. Scale bars 10 µm and 2 µm for whole cell image and magnified time-lapse frames, respectively. (**B**) Two representative kymographs of transient NMIIA pulses occurring at the ventral cortex of U2OS cells expressing eGFP-NMIIA, LifeAct-mKate 1.31 and miRFP670-paxillin. The 'non-productive' NMIIA pulses are accompanied by transient accumulation of F-actin, but without emergence of paxillin-positive nascent adhesions and stabilization of the actomyosin bundle. Kymographs were obtained from TIRF time-lapse videos with imaging interval of 10 s. (**C**) Two representative kymographs (obtained from *Video 7*) illustrating how emergence of a NMIIA pulse (green) leads to enforcement of adhesions (magenta) and maturation of a cortical stress fiber. White dashed lines in the kymographs indicate the onsets of NMIIA pulses. See also *Figure 3D*. (**D**) Percentual distribution of cortical stress fiber numbers in U2OS cells and mouse embryonic fibroblasts (MEFs) after different pharmacological perturbations. Cortical stress fiber number and length measurements were obtained through blind analysis of the tile-scan TIRF data. n = DMSO 157 and 247, CK-666, 166, and 182, Blebbistatin 195 and 198 cells for U2OS cells and MEFs, respectively. See also respective *Figure 4—figure supplement 2*A and C.

The online version of this article includes the following source data and figure supplement(s) for figure 4:

*Figure 4 continued on next page*

*Figure 4 continued*

**Source data 1.** Cortical SF number quantification after pharmacological inhibitions.
**Figure supplement 1.** Different ways to assemble cortical stress fibers from the ventral actomyosin cortex.
**Figure supplement 1—source data 1.** Cortical SF number quantification after formin inhibitions.
**Figure supplement 2.** Cortical stress fiber assembly is dependent on functional non-muscle myosin II (NMII) and integrin-based adhesions.

instead of a focal adhesion (*Figure 4—figure supplement 1A*, yellow arrow, and *Video 9*). Thus, generation of cortical stress fibers is both very dynamic and plastic process, and it can involve either *de novo* formation of focal adhesions or 'recycling' of pre-existing focal adhesions or higher-order actomyosin structures to connect the nascent cortical stress fiber to the cell cortex.

To examine the role of NMII in the assembly of cortical stress fibers more closely, we applied NMII inhibitor para-amino-Blebbistatin. As reported earlier (*Baird et al., 2017*), Blebbistatin did not cease the NMII pulsatile behavior in U2OS cells. Thus, myosin pulses do not depend on the catalytic activity of NMII, but can occur also if the motor domain of myosin is brought to low actin-affinity state by Blebbistatin (*Kovács et al., 2004*). Interestingly, the cortical stress fiber formation was almost completely abolished in Blebbistatin-treated cells, providing evidence that NMII activity is critical for the assembly of cortical stress fibers (*Figure 4—figure supplement 2A* and *Video 11*). We also cultured U2OS cells on poly-D-lysine coated imaging dishes to examine the importance of integrin-based adhesions in this process. By imaging cells that had managed to adhere to the dish only from their edges, but were devoid of adhesions inside cell margins, we revealed that while NMII pulses still drove transient cortical F-actin enrichment, they could not engage F-actin bundling to the assembly of cortical stress fibers (*Figure 4—figure supplement 2B* and *Video 12*).

We also applied the Arp2/3 inhibitor CK-666 to elucidate the possible role of Arp2/3 complex mediated actin filament assembly in the generation of cortical stress fibers. Even after 4 hr post drug administration, de novo cortical stress fiber assembly was frequently observed (*Figure 4—figure supplement 2C* and *Video 13*). Quantification of cortical stress fibers from control (DMSO-treated), para-amino-Blebbistatin, and CK-666 treated U2OS cells and MEFs confirmed that NMII inhibition led to an almost complete loss of cortical stress fibers, whereas Arp2/3 inhibition resulted in only a moderate decrease in the amount of these actomyosin bundles (*Figure 4D*).

Because formins were shown to contribute to assembly of dorsal stress fibers and to nucleation of actin filaments in the actin cortex (*Hotulainen and Lappalainen, 2006*; *Tojkander et al., 2011*; *Bovellan et al., 2014*; *Cao et al., 2020*), we examined the effects of formins on cortical stress fibers. We first used the SMIFH2 compound, which inhibits formins, but at higher concentration also increasingly affects the activities of various myosins (*Rizvi et al., 2009*; *Nishimura et al., 2020*). Treatment of U2OS cells for 1 hr with 12.5 µM SMIFH2 decreased the amount of cortical stress fibers, suggesting that formins contribute, either directly or indirectly, to the assembly of cortical stress fibers. However, subsequent siRNA experiments on mDia1, mDia2 and mDia3, from which mDia1-2 have been linked to assembly of dorsal stress fibers and transverse arcs, did not result in significant reduction in the number of cortical stress fibers (*Figure 4—figure supplement*

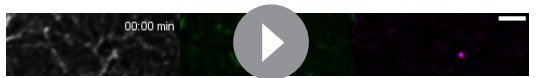

**Video 9.** NMIIA pulse-coordinated cortical stress fiber assembly in a migrating U2OS cell, where one end of the fiber connects to actomyosin mesh. Zoom in of a U2OS cell expressing LifeAct-TagGFP2 (gray), mApple-NMIIA (green), and miRFP670-paxillin (magenta), migrating on a fibronectin (10 µg/ml)-coated high-precision imaging dish. Related to *Figure 4—figure supplement 1A*. Time-lapse ring-TIRFM video was recorded with Deltavision OMX SR with 10s imaging interval. Playback rate 20 frames/s. Scale bar 2 µm.
https://elifesciences.org/articles/60710#video9

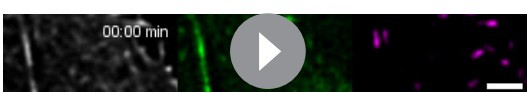

**Video 10.** NMIIA pulse-coordinated cortical stress fiber assembly in a migrating U2OS cell, where one end of the fiber connects to a pre-existing adhesion. Zoom in of a U2OS cell expressing LifeAct-TagGFP2 (gray), mApple-NMIIA (green), and miRFP670-paxillin, migrating on a fibronectin (10 µg/ml)-coated high-precision imaging dish. Related to *Figure 4—figure supplement 1B*. Time-lapse ring-TIRFM video was recorded with Deltavision OMX SR with 10s imaging interval. Playback rate 20 frames/s. Scale bar 2 µm.
https://elifesciences.org/articles/60710#video10

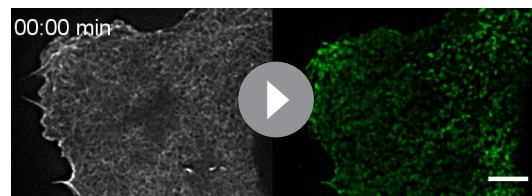

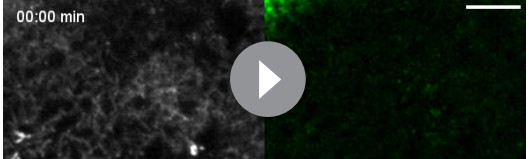

**Video 11.** Inhibiting non-muscle myosin II (NMII) activity abolishes cortical stress fiber assembly. Zoom in of a human osteosarcoma (U2OS) cell expressing LifeAct-TagGFP2 (gray), mApple-NMIIA (green), and miRFP670-paxillin (magenta, not shown in the zoom in) migrating on a fibronectin (10 µg/ml)-coated high precision imaging dish after 1 hr treatment with 50 µM p-amino-Blebbistatin. Related to **Figure 4—figure supplement 2A**. Time-lapse ring-TIRFM video was recorded with Deltavision OMX SR with 30 s imaging interval. Playback rate 20 frames/s. Scale bar 5 µm.
https://elifesciences.org/articles/60710#video11

**Video 12.** Inhibiting integrin-based ECM adhesion obstructs cortical stress fiber formation. Zoom in of a human osteosarcoma (U2OS) cell expressing LifeAct-TagGFP2 (gray), mApple-NMIIA (green), and miRFP670-paxillin (magenta, not shown in the zoom in) on a poly-D-lysine (10 µg/ml)-coated high-precision imaging dish. Imaging started 24 hr post plating. Related to **Figure 4—figure supplement 2B**. Time-lapse ring-TIRFM video was recorded with Deltavision OMX SR with 20 s imaging interval. Playback rate 20 frames/s. Scale bar 5 µm.
https://elifesciences.org/articles/60710#video12

**1C and D**). These results suggest that mDia1-3 are either not critical for generation of cortical stress fibers or that they are functionally redundant with each other or with some other formins expressed in U2OS cells.

Collectively, these results demonstrate that generation of cortical stress fibers is driven by NMII-catalyzed reorganization of the cortical actin filament network and dependent on integrin-based cell-matrix adhesions.

## Discussion

Here we discovered a new mechanism by which focal adhesion-attached stress fibers assemble in cells. We show that unlike ventral stress fibers, which are generated from the pre-exiting network of dorsal stress fibers and transverse arcs (**Tojkander et al., 2015**), cortical stress fibers located at the ventral cortex of 2D-migrating cells assemble *de novo* through NMII-dependent reorganization of the actin cortex (**Figure 5A and B**). It is important to note that although generation of cortical stress fibers below the nucleus and transverse arcs at the lamellipodium–lamellum interface display some parallels, transverse arc formation does not involve prominent myosin pulses, but they are rather assembled through condensation of the lamellipodial actin filament network during the retraction phase (**Burnette et al., 2011**). Cortical stress fibers bear close resemblance to the ventral stress fibers because they terminate to focal adhesions at both ends, do not elongate through actin polymerization at focal adhesions, and are composed of bipolar NMII filaments and α-actinin decorated actin filaments. However, they are thinner, more dynamic, devoid of NMII stacks, and less contractile compared to ventral stress fibers.

The assembly of cortical stress fibers is intimately linked to the actin cortex, where stochastic, NMII-mediated contractions of the cortical actin meshwork can lead to the formation of cortical actomyosin bundles. These are initially transient by nature and can either form a cortical stress fiber or collapse back into the ventral cortex. Although NMII-pulses have not been

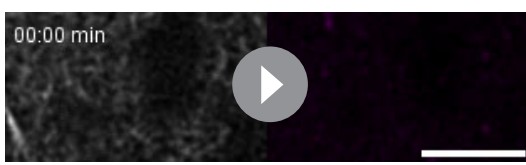

**Video 13.** CK-666 mediated Arp2/3 inhibition does not prevent cortical stress fiber assembly. Zoom in of a human osteosarcoma (U2OS) cell expressing LifeAct-TagGFP2 (gray) and miRFP670-paxillin (magenta) migrating on a fibronectin (10 µg/ml)-coated high-precision imaging dish after 4 hr treatment with 100 µM CK-666. Related to **Figure 4—figure supplement 2C**. Time-lapse ring-TIRFM video was recorded with Deltavision OMX SR with 30 s imaging interval. Playback rate 20 frames/s. Scale bar 5 µm.
https://elifesciences.org/articles/60710#video13

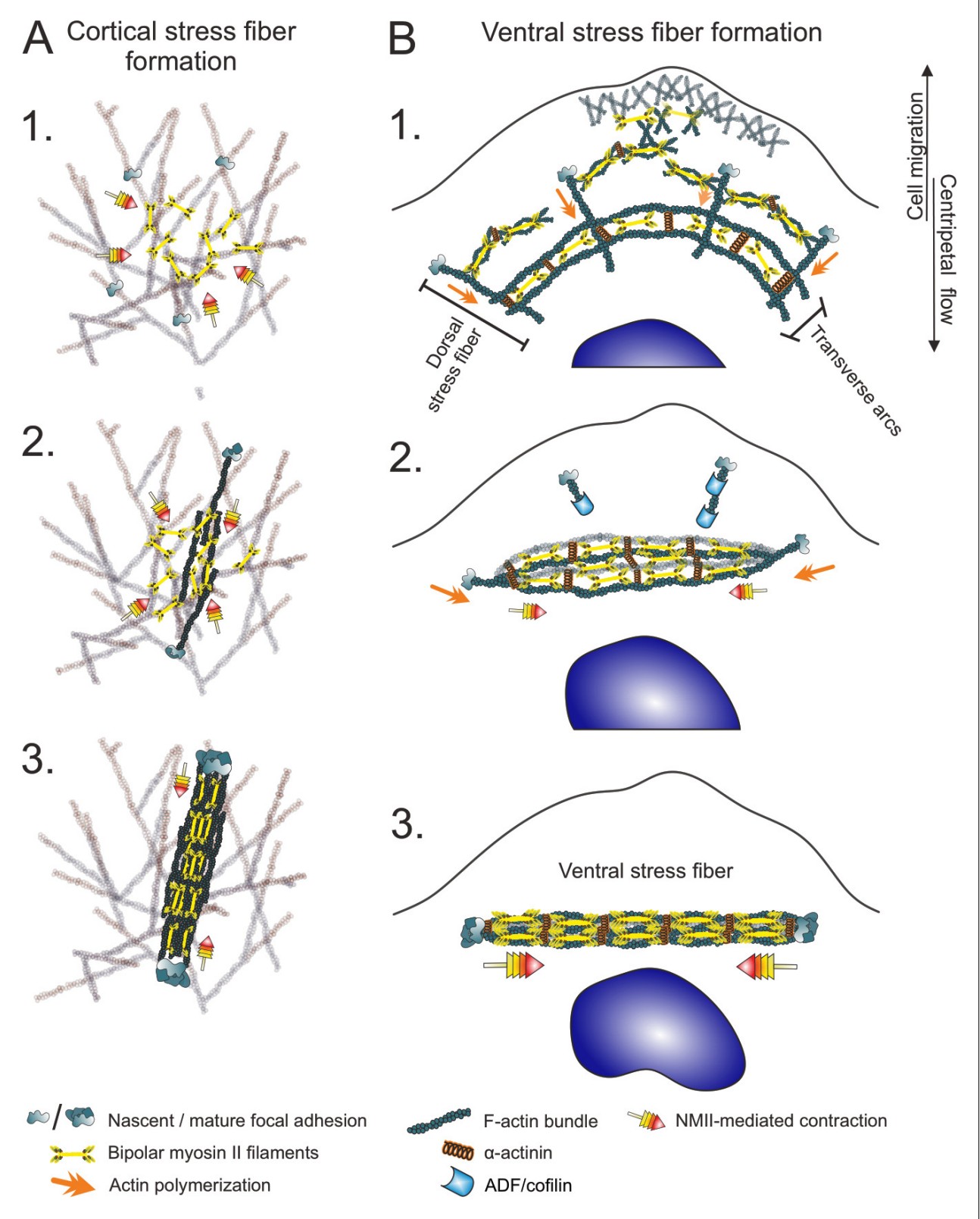

**A** Cortical stress fiber formation

1.

2.

3.

**B** Ventral stress fiber formation

1.

Dorsal stress fiber

Transverse arcs

Cell migration

Centripetal flow

2.

3.

Ventral stress fiber

Nascent / mature focal adhesion

Bipolar myosin II filaments

Actin polymerization

F-actin bundle

α-actinin

ADF/cofilin

NMII-mediated contraction

**Figure 5.** Two different mechanisms for generation of contractile stress fibers. (**A**) Schematic representation of the *de novo* assembly of a cortical stress fiber from the ventral actomyosin cortex. 1. Non-muscle myosin II (NMII) pulses occur frequently at the ventral actin cortex. 2. These pulses can cause transient accumulation and bundling of the cortical actin filaments via myosin-mediated actin filament crosslinking and reorganization. This triggers the enlargement of nascent focal adhesions at the ends of the bundle. 3. The transient actomyosin bundle can mature to a cortical SF through

*Figure 5 continued on next page*

*Figure 5 continued*

recruitment of more actin filaments and NMII. (**B**) Ventral stress fibers assemble from pre-existing stress fiber precursors at the lamellum of a migrating cell. 1. Dorsal stress fibers elongate through actin filament polymerization at focal adhesions, whereas transverse arcs are generated from NMII filaments and actin filaments at the lamellipodium–lamella interface. Assembly of transverse arcs does not involve prominent myosin pulses, but they are formed during retraction phase of lamellipodium (*Burnette et al., 2011*). The network of dorsal stress fibers and transverse arcs flows toward the cell center, and the arcs fuse with each other to form thicker actomyosin bundles. 2. The dorsal stress fibers in parallel to the transverse arcs are under strongest tension and begin to align with the fusing transverse arc network. On the other hand, the dorsal stress fibers arranged perpendicularly to the fusing transverse arcs sense lower tension and are disassembled by ADF/cofilins. 3. This leads to the formation of a thick stress fiber that is attached to focal adhesions at both ends (*Tojkander et al., 2015*; *Tojkander et al., 2018*).

previously linked to assembly of contractile actomyosin bundles, transient NMII pulse-driven cortical actin condensations were reported to occur in epithelial tissues of developing organisms where they drive tissue morphogenesis (*Munro et al., 2004*; *Solon et al., 2009*; *Rauzi et al., 2010*; *Munjal et al., 2015*; *Michaux et al., 2018*) and in cells of mesenchymal origin migrating in 2D environment (*Kim and Davidson, 2011*; *Nie et al., 2015*; *Baird et al., 2017*; *Graessl et al., 2017*). In *Drosophila* epithelia, the NMII/actin foci were also reported to undergo E-cadherin-mediated stabilization into medioapical filaments (*Mason et al., 2013*; *Vasquez et al., 2014*). Thus, myosin II pulse-driven F-actin bundling may have a more general role in the formation of different types of contractile actomyosin bundles in cells.

Also during cytokinesis, small actomyosin nodes can coalescence and form highly ordered contractile bundles at the cleavage furrow (*Laplante et al., 2016*; *Reymann et al., 2016*; *Henson et al., 2017*). Thus, initially very similar cortical actomyosin foci appear drive the formation of various different contractile structures. Similarly, the nascent cortical stress fibers come in many intermediate shapes that reflect the initially random formation of NMII/F-actin clusters. These intermediate structures can become momentarily engaged by more than two focal adhesions, but during subsequent tug-of-war-type process usually two focal adhesions become dominant at the ends of the cortical stress fiber. Moreover, a recent study reported emergence of new actomyosin bundles from the actin cortex close to a pre-existing stress fiber, and demonstrated that these actomyosin bundles exhibit a mechanical continuum with the actomyosin cortex (*Vignaud et al., 2020*). The degree of cortical F-actin network bundling may be dependent on both the mesh gap-size and thickness of the actin meshwork that vary between different cell types (*Bovellan et al., 2014*; *Svitkina, 2020*). Interestingly, the actin cortex appears to be more dense and under higher tension at the back of the 2D-migrating cells (*Chugh and Paluch, 2018*; *Bisaria et al., 2020*), where we observe the most prominent cortical stress fiber assembly.

What could be the reason underlying the assembly of cortical stress fibers under the nucleus in migrating mesenchymal cells? Transportation of the relatively rigid nucleus in cells migrating in tissue environment poses mechanical challenges, especially when facing narrow passages (*Petrie et al., 2014*; *Denais et al., 2016*). This results in accumulation of contractile F-actin bundles around the nucleus to mediate movement of the nucleus through the constriction (*Thomas et al., 2015*; *Davidson et al., 2020*). Similarly, when subjecting MEF or U2OS cells to adhere and migrate along narrow ECM-coated tracks, directional F-actin bundles are observed under the nucleus (*Doyle et al., 2012*; *Lee et al., 2018*). The stress fibers underneath the nucleus appear to be largely independent of linker of nucleus and cytoskeleton (LINC)-complex, which is known to mediate contacts between the nuclear envelope and perinuclear cap fibers at the dorsal surface of the nucleus (*Luxton et al., 2010*; *Kim et al., 2012*; *Kim et al., 2014*; *Calero-Cuenca et al., 2018*; *Lee et al., 2018*). Interestingly, RhoA pulses upstream of NMII were also found to localize mainly near the cell center (*Graessl et al., 2017*). Although the mechanisms by which these actomyosin bundles protecting the nucleus are assembled is still unknown, we envision that it would be difficult to generate such fibers from the transverse arc network, which to our knowledge is not generally present in cells migrating in a 3D environment (*Doyle et al., 2015*; *Thomas et al., 2015*). Thus, the *de novo* mechanism to generate contractile cortical stress fibers reported here is well suited for generation of stress fibers in 3D environment to protect the nucleus during migration. In line with this hypothesis, cells in 3D

environment utilizing mesenchymal migration mode exhibit multiple focal adhesion-mediated ECM attachments around their dorso-ventral and antero-posterior axis, and impose greater challenges with protecting the integrity of the nucleus than cells in 2D (*Yamada and Sixt, 2019*). We indeed observed short, adhesion-attached actomyosin bundles close to nuclei in U2OS cells migrating in 3D fibrin network. However, whether these structures correspond to cortical stress fibers and are generated from the actomyosin cortex through the same mechanism remain important questions for future research.

Taken together, we demonstrate that the myosin II pulse-induced actin contractions are linked to maturation of nascent focal adhesions at the cell cortex, and this can lead to an assembly of a contractile stress fiber that is attached to focal adhesions at its both ends. This new mechanism expands the toolbox that cells can apply for generation of stress fibers, thus allowing assembly of stress fibers also in conditions where the pre-existing stress fiber precursors (transverse arcs and dorsal stress fibers) are not present. In the future, it will be important to examine the functional relevance of cortical stress fibers especially in 3D environment, where transverse arcs are not present and focal adhesion formation is not restricted to the ventral plane.

# Materials and methods

## Key resources table

| Reagent type (species) or resource | Designation | Source or reference | Identifiers | Additional information |
|---|---|---|---|---|
| Cell line (*Homo sapiens*) | Human osteosarcoma cells (U2OS) | ATCC | Cat# HTB-96, RRID: CVCL_0042 | Authenticated through STR profiling |
| Cell line (*Mus musculus*) | Mouse embryonic fibroblasts (MEFs) | *Jiu et al., 2015* | | A gift from John Eriksson (University of Turku, Finland) |
| Recombinant DNA reagent | eGFP-NMIIA (plasmid) | Addgene, *Jacobelli et al., 2009* | RRID:Addgene_38297 | A gift from Mathew Krummel (University of California, San Francisco) |
| Recombinant DNA reagent | PA-GFP-Actin (plasmid) | Addgene | RRID:Addgene_57121 | A gift from Michael Davidson |
| Recombinant DNA reagent | LifeAct-mKate1.31 (plasmid) | Addgene, *Shcherbo et al., 2009* | RRID:Addgene_54668 | A gift from Michael Davidson |
| Recombinant DNA reagent | mApple-NMIIA (plasmid) | Addgene, *Burnette et al., 2014* | RRID:Addgene_54929 | A gift from Michael Davidson |
| Recombinant DNA reagent | Vinculin-mApple (plasmid) | Addgene | RRID:Addgene_54962 | A gift from Michael Davidson |
| Recombinant DNA reagent | α-actinin1-TagRFP-T (plasmid) | Addgene | RRID:Addgene_58033 | A gift from Michael Davidson |
| Recombinant DNA reagent | LifeAct-TagGFP2 (plasmid) | *Stefani et al., 2017* | | A gift from Emmanuel Lemichez (Pasteur Institute, France) |
| Recombinant DNA reagent | Histone H2B-eGFP (plasmid) | This paper | | A gift from Maria Vartiainen (University of Helsinki, Finland) |
| Recombinant DNA reagent | Histone H2B-mCherry (plasmid) | This paper | | A gift from Maria Vartiainen (University of Helsinki, Finland) |
| Recombinant DNA reagent | eGFP-Paxillin (plasmid) | This paper | | *Gallus gallus* paxillin ORF with T132S and M133L unattended mutations (pPL1277) |
| Recombinant DNA reagent | miRFP670-Paxillin (plasmid) | This paper | | eGFP in eGFP-Paxillin replaced with miRFP670-N1 (pPL1514) |
| Recombinant DNA reagent | miRFP670-N1 (plasmid) | Addgene, *Shcherbakova et al., 2016* | RRID:Addgene_79987 | A gift from Vladislav Verkusha (Albet Einstein College of Medicine, NY) |

*Continued on next page*

*Continued*

| Reagent type (species) or resource | Designation | Source or reference | Identifiers | Additional information |
|---|---|---|---|---|
| Transfected construct (human) | SMARTpoolON-TARGETplus siRNA to mDia1 | Dharmacon | Cat# L-010347-00-0005 | Transfected using GeneSilencer |
| Transfected construct (human) | SMARTpoolON-TARGETplus siRNA to mDia2 | Dharmacon | Cat# L-018997-00-0005 | Transfected using GeneSilencer |
| Transfected construct (human) | SMARTpoolON-TARGETplus siRNA to mDia3 | Dharmacon | Cat# L-012029-00-0005 | Transfected using GeneSilencer |
| Transfected construct (human) | negative (non-targeting) control siRNA | Qiagen | Cat# 1027281 | Transfected using GeneSilencer |
| Antibody | anti-vinculin (mouse monoclonal) | Sigma Aldrich | Cat# V9131, RRID:AB_477629 | 1:400 (IF) |
| Antibody | anti-NMIIA (C-termini) (rabbit polyclonal) | Biolegend | Cat# 909801, RRID:AB_2565100 | 1:800 (IF) |
| Antibody | anti-fibronectin (rabbit polyclonal) | Sigma Aldrich | Cat# F3648, RRID:AB_476976 | 1:400 (IF) a gift from Johanna Ivaska (University of Turku, Finland) |
| Antibody | anti-Y118-paxillin (rabbit monoclonal) | Cell Signaling Technology | Cat# 2541, RRID:AB_2174466 | 1:50 (IF) a gift from Johanna Ivaska (University of Turku, Finland) |
| Antibody | anti-human mDia1 (rabbit monoclonal) | Abcam | Cat# ab129167, RRID:AB_11143749 | 1:1000 (WB) |
| Antibody | anti-human mDia2 (rabbit polyclonal) | Proteintech | Cat# 14342-1-AP, RRID:AB_2092930 | 1: 500 (WB) |
| Antibody | anti- human mDia3 (rabbit polyclonal) | Sigma Aldrich | Cat# HPA005647, RRID:AB_1078657 | 1:10,00 (WB) |
| Antibody | anti-GAPDH (rabbit polyclonal) | Sigma Aldrich | Cat# G9545, RRID:AB_796208 | 1:10,000 (WB) |
| Antibody | Anti-mouse Alexa Fluor 488 (goat polyclonal) | Thermo Fisher Scientific | Cat# A-11001, RRID:AB_2534069 | 1:200 (IF) |
| Antibody | Anti-mouse Alexa Fluor 568 (goat polyclonal) | Thermo Fisher Scientific | Cat# A-11031, RRID:AB_144696 | 1:200 (IF) |
| Antibody | Anti-rabbit Alexa Fluor Plus 647 (goat polyclonal) | Thermo Fisher Scientific | Cat# A32733, RRID:AB_2633282 | 1:400 (IF) |
| Antibody | HRP-conjugated anti–rabbit (goat polyclonal) | Thermo Fisher Scientific | Cat# 32460, RRID:AB_1185567 | 1:1000 (WB) |
| Commercial assay or kit | Lipofectamine 2000 | Thermo Fisher Scientific | Cat# 11668019 | Plasmid transfection (3.5:1 DNA: Lipofectamine ratio) |
| Commercial assay or kit | GeneSilencer | Genlantis | Cat# T500750 | siRNA transfection |
| Commercial assay or kit | jetPRIME | Polyplus transfection | Cat# 114–01 | Plasmid transfection (TFM) |
| Commercial assay or kit | NEBuilder | New England Biolabs | Cat# E5520S | |
| Chemical compound, drug | Para-amino Blebbistatin | Optopharma | Cat# DR-Am-89 | 50 µM working concentration |
| Chemical compound, drug | SMIFH2 | Sigma Aldrich | Cat# S4826 | 12.5 µM working concentration |
| Chemical compound, drug | CK-666 | Sigma Aldrich | Cat# SML0006 | 100 µM working concentration |

*Continued on next page*

*Continued*

| Reagent type (species) or resource | Designation | Source or reference | Identifiers | Additional information |
|---|---|---|---|---|
| Chemical compound, drug | DMSO | Sigma Aldrich | Cat# D2650 | Diluent/negative control |
| Software, algorithm | Fiji | *Schindelin et al., 2012* | RRID:SCR_002285 | |
| Software, algorithm | OriginPro | OriginLab Corp. | RRID:SCR_014212 | |
| Software, algorithm | NMII pulse quantification (macro) | This paper. Modified from *Kim and Davidson, 2011* | | Available at: https://github.com/UH-LMU/lmu-users/tree/master/jaakko/NMIIA_pulses |
| Other | Fibrinogen (from human plasma) | Sigma Aldrich | Cat# 341578 | 2.25 mg/ml working concentration |
| Other | Thrombin (from human plasma) | Sigma Aldrich | Cat# T6884 | 1 U/ml working concentration |
| Other | Fibronectin (human plasma) | Sigma Aldrich | Cat# FIBRP-RO | 10 µg/ml working concentration |
| Other | Laminin | Sigma Aldrich | Cat# L2020 | 25 µg/ml working concentration |
| Other | Collagen | Sigma Aldrich | Cat# C4243 | 500 µg/ml working concentration |
| Other | poly-D-lysine | Sigma Aldrich | Cat# P0899 | 10 µg/ml working concentration |
| Other | DAPI stain | Thermo Fisher Scientific | Cat# D1306 | 0.2 µM working concentration (IF) |
| Other | Alexa Fluor 488-Phalloidin | Thermo Fisher Scientific | Cat# A12379 | 1:200 (IF) |
| Other | Alexa Fluor 568-Phalloidin | Thermo Fisher Scientific | Cat# 12380 | 1:200 (IF) |
| Other | Fluorescent microspheres. 0.2 µm diameter | Invitrogen | Cat# F8848 | |

## Cell culture and transfections

Wild-type MEFs (a gift from J.Eriksson's lab) and U2OS (ATCC HTB-96, verified through STR profiling) cells were cultured in high-glucose DMEM (Lonza, BE12-119F), supplemented with 10% FBS (Gibco, 10500–064), 10 U/ml penicillin, 10 µg/ml streptomycin, and 20 mM L-glutamine (Gibco, 10378–016) (from here on referred as complete DMEM) and kept at +37°C in a humidified atmosphere with 5% $CO_2$. Regular mycoplasma testing has been conducted with Lonza MycoAlert plus (LT07-710). For live-cell imaging experiments, Fluorobrite DMEM (Gibco, A1896701), supplemented with 25 mM HEPES and 10% FBS was used. Transient transfections were performed with Lipofectamine 2000 (Thermo Fisher Scientific) according to the manufacturer's instructions. 24 hr after transfection, cells were either fixed with 4% PFA in PBS for 10 min at +37°C (3D-SIM/tile-scan TIRF) or detached with 0.05% Trypsin-EDTA (Gibco, 15400054) and plated onto high-precision (#1.5H) 35 mm imaging dishes (Ibidi µ-dish high, 81158), coated with 10 µg/ml fibronectin (Sigma Aldrich, FIBRP-RO) for 1 hr RT (live-cell TIRF imaging experiments). To prevent integrin-mediated adhesion, 10 µg/ml poly-D-lysine (Sigma Aldrich, P0899) was applied onto the 35 mm imaging dishes at RT for 10 min, washed, and allowed to dry prior plating the cells. Cells were allowed to attach for 24 hr prior onset of imaging. For 3D-SIM or tile-scan TIRF, cells were plated onto #1.5H coverslips coated with 10 µg/ml fibronectin (Sigma Aldrich, FIBRP-RO), 25 µg/ml laminin (Sigma Aldrich, L2020), or 500 µg/ml collagen (Sigma Aldrich, C4243).

## Plasmids

eGFP-NMIIA was a gift from Mathew Krummel (Addgene plasmid #38297; *Jacobelli et al., 2009*). PA-GFP-Actin, LifeAct-mKate1.31, mApple-NMIIA, Vinculin-mApple, and α-actinin1-TagRFP-T were all gifts from Michael Davidson (Addgene plasmids #57121, #54668, #54929, #54962, #58033, respectively; *Burnette et al., 2014*; *Shcherbo et al., 2009*). LifeAct-TagGFP2 was kindly provided by the Emmanuel Lemichez lab. Histone H2B-eGFP (in peGFP-N3 backbone) and Histone H2B-mCherry (in pmCherry-N3 backbone) were gifts from Maria Vartiainen lab. miRFP670-Paxillin (pPL1514) was constructed by replacing the eGFP in eGFP-paxillin (pPL1277) (containing *Gallus gallus* paxillin ORF with T132S and M133L unattended mutations in an eGFP-C1 backbone) with miRFP670 originated from miRFP670-N1 (a gift from Vladislav Verkhusha, Addgene plasmid #79987; *Shcherbakova et al., 2016*) via restriction free cloning using NEBuilder (New England Biolabs). All inserts have been sequence verified prior the respective plasmid usage.

## Reagents

Mouse monoclonal antibody (ab) for vinculin (Sigma Aldrich, V9131), rabbit polyclonal ab recognizing the C-terminal tail of the NMIIA heavy chain (Biolegend, 909801), rabbit polyclonal anti-fibronectin (F3648, Sigma Aldrich), and monoclonal rabbit ab against Y118-phosphorylated paxillin (#2541, Cell Signaling Technology) were employed in this study. 4′,6′-diamidino-2-phenylindole (DAPI) (Thermo Fisher Scientific, D1306) was utilized to detect the nuclei, whereas Alexa Fluor 488- and 568-Phalloidin (Thermo Fisher Scientific, A12379/12380) were applied to visualize F-actin. Alexa Fluor 488- and 568-conjugated goat anti-mouse (Thermo Fisher Scientific, A-11001 and A-11031, respectively) and Alexa Fluor Plus 647-conjugated goat anti-rabbit (Thermo Fisher Scientific, A32733) were used as secondary antibodies. 5% BSA-TBS-Tween20 (0.02%) was used as a blocking buffer for the cells prior staining as well as a diluent for the abovementioned reagents. Monoclonal rabbit ab against human mDia1 (Abcam, ab129167) and polyclonal rabbit abs raised against human mDia2 (Proteintech, 14342-1-AP) and human mDia3 (Sigma Aldrich, HPA005647) were used in 1:500 and 1:1000 dilution, respectively, in 5% milk TBS-Tween20 (0.05%) to detect siRNA depletion efficiency. Polyclonal rabbit ab against GAPDH (Sigma-Aldrich, G9545) was used in 1:10,000 dilution to probe equal loading in WB. 50 µM of para-amino-Blebbistatin (Optopharma, DR-Am-89) was applied for 1 hr to inhibit NMII activity, 100 µM CK-666 (Sigma Aldrich, SML0006) for 4 hr to inhibit the Arp2/3 complex, and 12.5 µM of SMIFH2 (Sigma Aldrich, S4826) was added for 1 hr to inhibit formin activity. Inhibitor stocks were made to dimethyl sulfoxide (DMSO) (Sigma Aldrich, D2650) that was also used as control in the corresponding experiments. To study the role of mDia, mDia2 and mDia3 formins in the process, we treated U2OS cells for 72 hours with specific siRNAs (20 µM of SMARTpool ON-TARGETplus, L-010347-00-0005, L-018997-00-0005 and L-012029-00-0005 respectively, Dharmacon) or with 20 µM of negative control siRNA (Qiagen, 1027281). For 3D-SIM experiments, samples were mounted using either non-hardening Vectashield (Vector laboratories, H-1000) or Prolong Glass (Thermo Fisher Scientific, P36980). Fibrinogen (Sigma-Aldrich, 341578) and Thrombin (Sigma-Aldrich, T6884) were used to prepare the 3D-fibrin gels.

## 3D-SIM

All 3D-SIM images were obtained at RT, using Deltavision OMX SR (Cytiva) with 60×/1.42NA PlanApo N oil objective with 1.516 RI immersion oil, a laser module with 405-, 488-, 568-, and 640 nm diode laser lines and three sCMOS cameras, operated through Acquire SR 4.4 acquisition software. SI reconstruction and image alignment were performed with SoftWoRx 7.0. Imaging arrays of 1024 × 1024 or 512 × 512 were used, both with pixel size of 0.08 µm and 0.125 µm (x/y and z). Samples for 3D-SIM were prepared according to *Kraus et al., 2017* with exceptions of using 5% BSA-PBST as blocking reagent, and omitting the pre-incubation with mounting reagent when using the Prolong Glass as a mountant.

## TIRF microscopy

For live-cell TIRF experiments, the Ring-TIRF module from the Deltavision OMX SR (Cytiva) with 60×/1.49NA Apo N oil objective (Olympus), 1.522 RI immersion oil and imaging chamber with controlled humidified atmosphere of 5% $CO_2$ and 37°C was utilized. Sample illumination with 488, 568, and 640 nm diode lasers was detected and recorded with three respective sCMOS cameras and

controlled through Acquire SR 4.4 acquisition software. The captured 1024 × 1024 time-lapse videos had a pixel size of 0.08 μm (x/y). Photoactivation experiments were also performed with Deltavision OMX SR. There, 4% laser power (405 nm) in EPI mode with 1 ms exposure was applied to activate PA-GFP-Actin on a cortical stress fiber visualized by mKate1.31-LifeAct adjacent to a miRFP670-paxillin positive adhesion. The time-lapse imaging with 10 s interval for a total of 5 min was conducted in TIRF (cortical stress fiber) or in Highly Inclined and Laminated Optical sheet (HILO, dorsal stress fiber) mode by recording also three frames prior to photoactivation. Obtained time-lapse series were deconvolved and channels aligned with SoftWoRx 7.0. Prior the onset of live-cell imaging, cells were allowed to settle within the imaging chamber for 1 hr.

'Tile-scan' TIRF of fixed specimens on 35 mm imaging dishes (Ibidi μ-dish high) in 1× PBS was performed with Eclipse Ti-E N-STOM/TIRF microscope (Nikon Instruments) using 100×/1.49NA Apo TIRF oil objective. Illumination was provided by 405 nm (100 mW), 488 nm (Argon), and 561 nm (150 mW) laser lines and LU4A laser unit (Nikon Instruments), controlled via NIS-Elements (NIS-AR version 4.5). Images were captured using iXon+ 885 EMCCD camera (Andor Technology) with imaging array of 1004 × 1002 pixels, and final pixel size of 0.08 μm (x/y). In order to acquire the data set from the specimen in unbiased manner, focus was set, followed by capturing 5 × 5 field of view (FOV) tile-scan with 10% overlap. After this, the next tile-scan was acquired from a new area, six times the FOV to another direction. Tile-scans of mDia2 siRNA- and respective ctrl siRNA- treated samples were performed with Ring-TIRF module of Deltavision OMX SR, as for live-cell TIRF experiments but with following exceptions: imaging was performed at RT, using 1.518 RI immersion oil and 405, 488 and 640 nm diode lasers. 5x5 FOVs (1024 x 1024) including 10% overlap, were captured manually, followed by moving to a new area, at least over six times the FOV to another direction, at random on the imaging dish.

## Traction force microscopy (TFM)

For the TFM experiments, U2OS cells on a 35 mm imaging dishes (Mat-Tek, P35G-1.5–14C) were transfected with mKate1.31-LifeAct using JetPRIME transfection reagent (Polyplus transfection, 114–01) according to the manufacturer's instructions. Next day, cells were re-plated on polyacrylamide (PAA) gels of known stiffness (Young's Modulus/elastic modulus = 26 kPa) coated with mixture of collagen and fibronectin and incubated for 2–4 hr prior to imaging. PAA substrates were surface-coated with sulfate fluorescent microspheres of 0.2 μm diameter (Invitrogen, F8848) before coating with the ECM proteins. Cells and the underlying microspheres were imaged with 3I-Marianas imaging setup containing a heated sample chamber (+37°C) and controlled 5% $CO_2$ (3I intelligent Imaging Innovations, Germany). 63×/1.2 W C-Apochromat Corr WD = 0.28 M27 objective was used. After first round of images, the cells were removed from the substrates with 10× Trypsin (Lonza, BE02-007E) and a second set of images were obtained of the microspheres in a cell-free configuration. Microsphere displacement maps were achieved by comparing the first and second set of microsphere images. By knowing the spatial displacement field, substrate stiffness (26 kPa), and tracing manually cell boundaries and single adhesions in the ends of stress fibers, we could compute the traction fields by using Fourier Transform Traction Cytometry (*Tolić-Nørrelykke et al., 2002*; *Krishnan et al., 2009*). From the traction fields, root mean squared (RMS) magnitudes were computed. Importantly, several of the cortical stress fibers relayed forces so low that they were undetectable by the microsphere displacement assay, and thus the actual RMS values for this stress fiber subtype are likely to be even lower. Distinct stress fiber types for the measurements were chosen based on their appearance, location, and connections to FAs: Ventral stress fibers were defined as thick, straight bundles, usually behind an arc network and connected to focal adhesions from both ends; Cortical stress fibers were defined as thin and short bundles, usually located under the nucleus and associated with focal adhesions from both ends.

## Fibrin gel preparation and imaging

For the visualization of the actomyosin cytoskeleton in 3D environment, U2OS cells transfected with eGFP-Paxillin were embedded into a fibrin matrix according to protocol by *Owen et al., 2017* with small modifications. Briefly 15 mg/ml fibrinogen stocks were mixed with HBSS (Gibco, 14175095) to total volume of 800 μl. Subsequently, 100 μl of 9× DMEM supplemented with 10% FBS and 100 μl of 2.5 × 10⁵ cells/ml in complete DMEM were added, and mixed thoroughly, avoiding air bubble

formation to obtain final fibrinogen concentration of 2.25 mg/ml. One unit of Thrombin was added, the suspension was mixed by pipetting three times, and 300 µl aliquots were immediately transferred onto the 35 mm imaging dishes (Ibidi µ-dish high). Dishes were inverted and kept 15 min at +37˚C to allow fibrin polymerization. After polymerization, 3 ml complete DMEM was added and cells were allowed to spread for 48 hr prior fixation and staining.

The eGFP-paxillin expressing cells in fibrin gels were further stained with phalloidin and DAPI. The samples were imaged in 1× PBS with Andor Dragonfly 505 (Oxford Instruments) spinning disk confocal, using 60×/1.2NA Plan APO VC water immersion objective with 0.29 mm working distance. Samples were illuminated with 404 (100 mW), 488 (150 mW), and 561 nm (100 mW) direct modulation lasers, using 40 µm pinhole diameter and 1× camera magnification. 3D stacks using 190 nm step size, 2047 × 2047 resolution, and 0.1 µm pixel size (x/y) were captured with Andor Zyla 4.2 sCMOS camera controlled through Fusion 2.0 acquisition program. Acquired 3D stacks were further deconvoluted with the ClearVieW-GPU (Oxford Instruments). 3D-rendered images and videos were constructed using Bitplane Imaris 9.2.1 (Oxford Instruments).

## NMIIA pulse quantification

A Fiji/ImageJ macro originally used to quantify F-actin dynamics was obtained from Lance Davidson (*Kim and Davidson, 2011*), and used here to detect and track the frequency of NMIIA pulses. The macro was adapted to current file type and modified to save intermediate results which were further processed with a Python script. Pulses were tracked at the ventral cortex from TIRF time-lapse series of U2OS cells expressing mApple-NMIIA and H2B-eGFP, and analyzed similar to *Baird et al., 2017* with exceptions of defining the initial regions of interest (ROIs) so that pulses were recorded first from the ventral cortex excluding nucleus ('outside') and then only within the nucleus perimeter ('within'). Also more stringent 1.3 threshold of NMIIA signal over the background intensity was used and number of connected hexagons smaller than 3 (in individual x–y frames) were filtered out to improve the analysis by excluding single bright myosin puncta. Furthermore, the 'outside' ROI excluded edges of the cell as well as the early lamella, to prevent the NMII signal from these locations to bias the analysis. To minimize the effect of photobleaching, pulse frequency was determined by registering all pulses within the first 30 min of TIRFM time-lapse videos that persisted more than three frames. The frequency of each 'outside' data point was further normalized to the smaller ventral cortex area covered by the nucleus as calculated from the drawn ROIs with Fiji/ImageJ (*Schindelin et al., 2012*).

## Image processing and statistical analyses

All TIRFM data, excluding the LifeAct channel, was processed with the rolling ball background subtraction plug-in using a 50-pixel ball radius in Fiji/ImageJ. Temporal color-coded hyperstacks (3D-SIM) and kymographs (TIRF time-lapse data, KymographBuilder plugin) were also created with Fiji. The statistical analysis for TFM data, NMII pulse frequency, and cortical stress fiber localization, as well as bar graph generation, were performed with Excel (Microsoft). Half-violin plots were constructed with OriginPro 2020 (OriginLab Corp.). Statistical significance for myosin pulse frequency was calculated with paired *t*-test as respective 'inside' vs. 'outside' pulses were quantified within the same cell, preceded by examining data normality with quantile–quantile plot. To assess statistical difference for the TFM data, nonparametric Mann–Whitney *U*-test was applied. In quantification of cortical stress fiber formation from the TIRFM time-lapse data (*Figure 3B*), cortical stress fibers of maximum 10 µm diameter, persisting for 2 min or longer, having focal adhesion at both ends with possibility of sharing a common focal adhesion, were included in the analysis. To be categorized as forming under the nucleus, at least one focal adhesion had to reside below the nucleus. Number of cells analyzed for each quantification are listed in the respective figure legends. The cortical stress fibers' number from TIRF images of fixed cells depleted of mDia1/2 or treated with SMIFH2, Blebbistatin, CK-666, or DMSO alone were quantified by manual blind analysis. Here, the cortical stress fibers were defined as focal adhesion attached (from both ends) actin bundles located at least partially underneath the nucleus and with a length smaller than the diameter of nucleus.

## Acknowledgements

We thank the Biomedicum Imaging Unit of the University of Helsinki, sponsored by HiLIFE and Biocenter Finland, for support with imaging. Maria Vartiainen (University of Helsinki) and Johanna Ivaska (University of Turku) are acknowledged for providing reagents. We thank Lance Davidson (University of Pittsburgh) for providing the macro and Harri Jäälinoja (LMU imaging unit, Institute of Biotechnology, University of Helsinki) for help with modifying the macro for pulse quantification. This work was supported by grants from Sigrid Juselius Foundation (to PL), Jane and Aatos Erkko Foundation (to PL and ST), and Academy of Finland (to ST).

## Additional information

### Competing interests

Pekka Lappalainen: Reviewing editor, *eLife*. The other authors declare that no competing interests exist.

### Funding

| Funder | Grant reference number | Author |
| --- | --- | --- |
| Sigrid Juselius Foundation | 4708344 | Pekka Lappalainen |
| Aatos Erkko Foundation | 4704407 | Pekka Lappalainen |
| Academy of Finland | 294174 | Sari Tojkander |

The funders had no role in study design, data collection and interpretation, or the decision to submit the work for publication.

### Author contributions

Jaakko I Lehtimäki, Conceptualization, Investigation, Visualization, Methodology, Writing - original draft, Writing - review and editing, JL conducted all experiments and analyzed the data, expect for the TFM experiments; Eeva Kaisa Rajakylä, Investigation, Methodology, Conducted the TFM experiments; Sari Tojkander, Investigation, Methodology, Writing - review and editing, Conducted the TFM experiments; Pekka Lappalainen, Conceptualization, Funding acquisition, Writing - original draft, Project administration, Writing - review and editing

### Author ORCIDs

Jaakko I Lehtimäki (iD) https://orcid.org/0000-0002-0002-0242
Pekka Lappalainen (iD) https://orcid.org/0000-0001-6227-0354

### Decision letter and Author response

Decision letter https://doi.org/10.7554/eLife.60710.sa1
Author response https://doi.org/10.7554/eLife.60710.sa2

## Additional files

### Supplementary files

• Transparent reporting form

### Data availability

All data generated or analysed during this study are included in the manuscript and supporting files. Source data files for pulse quantification are provided in GitHub (https://github.com/UH-LMU/lmu-users/tree/master/jaakko/NMIIA_pulses).

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
