## [Decision Letter]

**Acceptance summary:**

This elegant study uses cutting-edge imaging techniques to reveal the existence of cortical stress fibers that assemble in the cortex underneath the nucleus. The characterization of these cellular structures improves our understanding of the mechanisms by which cells produce and sense force.

**Decision letter after peer review:**

Thank you for submitting your article "Generation of stress fibers through myosin-driven re-organization of the actin cortex" for consideration by *eLife*. Your article has been reviewed by three peer reviewers, including Reinhard Fässler as the Reviewing Editor and Reviewer #1, and the evaluation has been overseen by Anna Akhmanova as the Senior Editor. The following individual involved in review of your submission has agreed to reveal their identity: Frank Schnorrer (Reviewer #2). The reviewers have discussed the reviews with one another and the Reviewing Editor has drafted this decision to help you prepare a revised submission.

Summary:

The authors identified and characterised a novel type of stress fibre, which forms underneath the nucleus from the pre-existing cortical actin. The assembly of this new stress fibre termed cortical stress fiber is triggered by non-muscle myosin II pulses, which reorganise the cortical actin. The cortical stress fibers attach to and reinforce nascent adhesions.

Essential revisions:

1) The formin mDia1 and mDia2 are involved in stress fiber formation (Hotulainen and Lappalainen, 2006; Tojkander et al., 2011), and recently, mDia1 has been involved in the nucleation of actin filaments in the actin cortex (https://doi.org/10.1038/s41556-020-0531-y). The role of formins in the assembly of the cortical stress fibers with either SMIFH2 treatments or by depleting formins using siRNA or Crispr should be tested.

2) It is not clear whether the cortical stress fibers elongate through actin polymerization. FRAP experiment on cortical stress fibers in GFP-actin expressing cells (used by the authors in previous articles) would help clarifying this issue.

3) The sentence “Blebbistatin did not cease the NMII pulsatile behavior in U2OS cells. However, the cortical stress fiber formation was almost completely abolished, providing evidence that NMII activity is critical for the assembly of cortical stress fibers” is ambiguous. It is necessary to clarify the relationship between myosin pulses and mysoin motor activity. It is also important to illustrate the spatiotemporal correlation between NMII pulses and focal adhesion reinforcement (Figure 3D) and F-actin bundling (Figure 4A). NMII pulses are not obvious in in Figures 3D and 4A. We suggest to perform two-color (NMII/vinculin and NMII/actin) kymographs of cross-sections on existing time-lapses.

4) The authors should clarify the fate of cortical stress fibers (do they transit or mature to “ventral stress fibers”, or just disappear) and determine their average half-life.

5) The investigations should be complemented with 3D migrating cells. Particularly, since the authors rightly speculate that CSFs may protect the integrity of the nucleus in 3D migrating cells. Hence, they should also exist in 3D.

---

## [Author Response]

Essential revisions:1) The formin mDia1 and mDia2 are involved in stress fiber formation (Hotulainen and Lappalainen, 2006; Tojkander et al., 2011), and recently, mDia1 has been involved in the nucleation of actin filaments in the actin cortex (https://doi.org/10.1038/s41556-020-0531-y). The role of formins in the assembly of the cortical stress fibers with either SMIFH2 treatments or by depleting formins using siRNA or Crispr should be tested.

This is an excellent suggestion, and we have accordingly examined the involvement of formins in cortical stress fiber formation by using both SMIFH2 and siRNA. Experiments on SMIFH2 were performed with low concentration of the compound, because SMIFH2 was recently shown to also inhibit various myosins (Nishimura et al., 2020).

Nevertheless, the SMIFH2 experiments provided evidence that formins may indeed contribute to the assembly of cortical stress fibers. However, our siRNA experiments on Dia1 and Dia2 did not result in significant reduction in the number of cortical stress fibers, suggesting that these specific formins are either not critical for generation of cortical stress fibers, or that they are functionally redundant with each other or with some other formins expressed in U2OS cells. The new data are shown in Figure 4—figure supplement 1C and D, and discussed in the Results (and in the Materials and methods).

Concerning the role of formins in cortical stress fiber assembly, is important to note that, unlike dorsal stress fibers (e.g. Lappalainen and Hotulainen, 2006), cortical stress fibers do not elongate through actin polymerization at adhesions (see point #2 below). Therefore, formins most likely contribute to cortical stress fiber assembly indirectly, for example by nucleating actin filaments for the actin cortex.

2) It is not clear whether the cortical stress fibers elongate through actin polymerization. FRAP experiment on cortical stress fibers in GFP-actin expressing cells (used by the authors in previous articles) would help clarifying this issue.

We have now performed photoactivation experiments to examine possible elongation of cortical stress fibers through actin polymerization. These experiments (shown in Figure 2—figure supplement 1A, B, and discussed in the Results, as well as in the Materials and methods) revealed that cortical stress fibers do not elongate through actin polymerization at focal adhesions. This is also consistent with their assembly mechanism, which involves reorganization of the pre-existing cortical actin filament network rather than generation of new actin filaments.

3) The sentence “Blebbistatin did not cease the NMII pulsatile behavior in U2OS cells. However, the cortical stress fiber formation was almost completely abolished, providing evidence that NMII activity is critical for the assembly of cortical stress fibers” is ambiguous. It is necessary to clarify the relationship between myosin pulses and mysoin motor activity. It is also important to illustrate the spatiotemporal correlation between NMII pulses and focal adhesion reinforcement (Figure 3D) and F-actin bundling (Figure 4A). NMII pulses are not obvious in in Figures 3D and 4A. We suggest to perform two-color (NMII/vinculin and NMII/actin) kymographs of cross-sections on existing time-lapses.

We have now prepared kymographs of both “non-productive” and “productive” myosin pulses to illustrate the correlation between myosin pulses and adhesion reinforcement. These new data are shown in Figure 4B and C, and discussed in –the Results. We have also revised the manuscript text –Results to clarify the relationships between NMII pulses and NMII activity (i.e. that as described before by Baird et al., myosin pulses do not depend on the catalytic activity of NMII, but can occur also if the motor domain of myosin is inhibited).

4) The authors should clarify the fate of cortical stress fibers (do they transit or mature to “ventral stress fibers”, or just disappear) and determine their average half-life.

Although we occasionally observed fusion of cortical stress fibers with each other, they do not mature to thick, long “ventral” ventral stress fibers. Instead, majority of cortical stress fibers persist only a relative short period of time. This is illustrated by an analysis of longer (120 min) videos (see Author response image 1), which demonstrated that the half-life of cortical stress fibers is ~25 min. It is, however, important to note that a large variation exists between the lifetimes of individual cortical stress fibers. We now discuss the fate and half-life of cortical stress fibers in the Results.

**Author response image 1. respfig1:** Analysis of cortical stress fiber half-life in U2OS cells migrating on a fibronectin and expressing LifeAct-TagGFP2 and mApple-Paxillin. 120 min TIRFM time-lapse videos with 1-minute imaging interval. N = 15 cells; 77 cortical stress fibers. Please note that all stress fibers that were visible already in the beginning of videos and/or were present in the end of videos were excluded from the analysis. We only analysed cortical stress fibers of max 10 µm length arising at the cell center, approximately within the nucleus perimeter. It is also important to note that in many cases cortical stress fibers fused with each other, making it difficult to accurately measure their half-lifes.

5) The investigations should be complemented with 3D migrating cells. Particularly, since the authors rightly speculate that CSFs may protect the integrity of the nucleus in 3D migrating cells. Hence, they should also exist in 3D.

This an interesting point, and we have now examined the distributions of adhesions, actin filaments, and nuclei in U2OS cells migrating in 3D fibrin network. In addition to thick, long actin filament bundles, which we assume are related to “ventral stress fibers” observed in cells on 2D matrix, these cells also exhibited close to their nuclei relatively short and thin actin filament bundles, which terminated to paxillin-rich adhesions at their both ends. These new data are shown in Figure 3—figure supplement 1B and Video 6, and discussed in the Results, Discussion, as well as in the Materials and methods.

We also acknowledge that in order to confirm that these actin filament bundles indeed correspond to “cortical stress fibers”, one should carefully examine the assembly mechanisms of these actin filament bundles in a 3D environment. However, high-resolution, high-speed live-cell imaging in 3D is extremely challenging due to technical reasons, and thus we sincerely feel that such experiments are well beyond the scope of present study.